# Multi-Hazard and Spatial Transferability of a CNN for Automated Building Damage Assessment

**Tinka Valentijn** [1,2,*] **, Jacopo Margutti** [1] **, Marc van den Homberg** [1] **and Jorma Laaksonen** [2]

[1]  510, An Initiative of the Netherlands Red Cross, 2593 HT The Hague, The Netherlands;
    jmargutti@redcross.nl (J.M.); mvandenhomberg@redcross.nl (M.v.d.H.)
[2]  School of Science, Aalto University, 02150 Espoo, Finland; jorma.laaksonen@aalto.fi
[*]  Correspondence: tinka.valentijn@aalto.fi

**Abstract:** Automated classification of building damage in remote sensing images enables the rapid and spatially extensive assessment of the impact of natural hazards, thus speeding up emergency response efforts. Convolutional neural networks (CNNs) can reach good performance on such a task in experimental settings. How CNNs perform when applied under operational emergency conditions, with unseen data and time constraints, is not well studied. This study focuses on the applicability of a CNN-based model in such scenarios. We performed experiments on 13 disasters that differ in natural hazard type, geographical location, and image parameters. The types of natural hazards were hurricanes, tornadoes, floods, tsunamis, and volcanic eruptions, which struck across North America, Central America, and Asia. We used 175,289 buildings from the xBD dataset, which contains human-annotated multiclass damage labels on high-resolution satellite imagery with red, green, and blue (RGB) bands. First, our experiments showed that the performance in terms of area under the curve does not correlate with the type of natural hazard, geographical region, and satellite parameters such as the off-nadir angle. Second, while performance differed highly between occurrences of disasters, our model still reached a high level of performance without using any labeled data of the test disaster during training. This provides the first evidence that such a model can be effectively applied under operational conditions, where labeled damage data of the disaster cannot be available timely and thus model (re-)training is not an option.

**Keywords:** damage assessment; building damage; CNN; transfer learning; satellite

## 1. Introduction

Disasters caused by natural hazards affected 95 million people in 2019 alone and led to 140 billion USD in damage [1]. Providing adequate emergency response is increasingly challenging, especially in developing countries, due to the growing number of disasters and insufficient resources to provide humanitarian services [2]. High-quality information on the impact of disasters is essential [3]. It can be used in the immediate response phase to identify areas and people affected as well as in the mitigation phase to improve urban planning, for example. The impact of a disaster varies from tangible to intangible and from economic loss (such as of livelihoods in terms of crop and livestock) to physical damage to buildings and critical infrastructures. Different approaches, methodologies, and tools are used to collect impact data. Damage and needs assessments (DNAs) are systematic processes for determining the location, nature, and severity of the damage and they are usually done at various intervals from right after a disaster hits into the recovery phase [4–6]. DNAs typically rely on field surveys, complemented in some cases by, for example, aerial imagery. Field surveys are costly, are time-consuming, and are typically only possible once the affected regions are accessible, causing a delay in their availability. Additionally, they cannot be updated easily and are of varying quality. Damage

assessments using remotely sensed data are already often included in DNAs, if suitable imagery is available. These assessments are currently done manually, either by relief workers themselves or by specialized agencies such as UNOSAT. Different protocols for annotating damage in such images exist [7–9], which unfortunately prevents the comparison of assessments across different disasters or even within the same disaster, when carried out by different agencies. Additionally, the quality of these assessments is not consistent and can be insufficient [10], while their speed and scalability is limited (less than 100 buildings per hour per person).

To overcome these limitations, scholars and practitioners are increasingly interested in automated damage assessment using remotely sensed data [5,11]. Automated damage assessment has the potential to provide a more accurate, quicker, cheaper, and safer assessment than when done manually, thereby increasing the speed and cost-effectiveness of the disaster response. Remotely sensed data are especially suitable for the identification of physical damage to buildings, as this can be more easily identified than economic loss, and have also been successfully employed for social, economic, and environmental analysis [12]. Additionally, remotely sensed data from both satellites and unmanned aerial vehicles (UAV) have become progressively more available in recent years. Satellite data have the advantage of having high spatial coverage and being collected automatically, while UAVs can capture data at a higher resolution and without cloud cover.

Automated building damage classification using remote sensing can be implemented in the disaster response. It is likely to be most useful in the immediate response phase, i.e., within the 24 to 168 h period after the disaster, when DNAs are still not available. Automated damage assessments can be delivered very fast, within a few hours from when the remotely-sensed images become available. In Figure 1, we outline the DNA workflow and how an automated building damage assessment, in the form of a standalone tool, can enter into the picture. Once a disaster strikes (top left), the first task is to execute a rapid DNA to understand the scale of the damage caused by the disaster. For this, both primary (e.g., field surveys) and secondary (e.g., demographic or socio-economic) data are currently used. Damage assessments are part of this rapid DNA, and can potentially be provided automatically; the only action needed by the emergency response authorities is the request of adequate satellite images, either directly to one provider—many already provide this type of support—or through the International Charter: Space and Major Disasters. If open-source building footprints are not available, a building detection task (shown with dashed lines in Figure 1) has to be added. Model re-training, potentially using a small set of manually labeled data, is another additional process step. Since the labeling task is likely to be time-consuming, as explained before, this could enter as a refinement of the initial damage assessment and thus feed into the next DNAs or be used to optimize the model for a future disaster.

Research on automated building damage assessment using remote sensing imagery is still in its early stages [13]. Nex et al. pointed out in [13] that "visible structural building damages can look completely different depending on the disaster type considered, and a general approach that is automatically and accurately able to cope with all different typologies of disasters is not in reach at the moment". To be of use in emergency response operations, the performance of the classification model needs to be sufficiently high on an unseen disaster, i.e., a disaster for which no damage data are available, since using ground-sourced data to re-train the model invalidates most the advantages of this solution (speed, cost, and consistency).

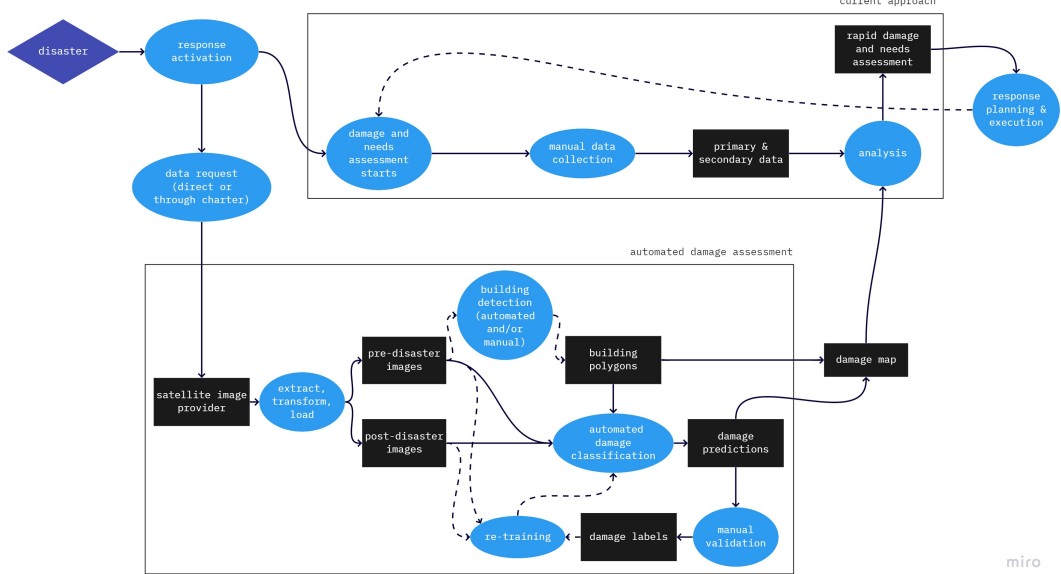

**Figure 1.** Workflow of DNAs during emergency response operations and the possible role of an automated building damage classification model. Black: items/data; blue: processes; purple: events. Full lines: necessary steps; dashed lines: optional steps.

At the same time, the model's variation in performance across disaster types and geographical locations needs to be quantifiable, so as to assess whether it can reliably be used in a new context. To date, however, few studies have examined the performance across disaster types or tested the performance on only the data that are available in real situations. The contribution of this research is to bridge this evidence gap, by assessing the multi-hazard and spatial transferability of an automated building damage classification model, and thus establishing if such a model can be used in emergency response operations. Understanding these aspects of transferability is essential for working towards an automated damage assessment that fits in the flow that is sketched in Figure 1. The model proposed in this research builds on previous work and was designed with data requirements and transferability in mind. Based on previous research, discussed in Section 2, a CNN-based model is the most promising candidate. The CNN model used in this research first separately extracts features from imagery before and after the disaster, and then those separate features are concatenated. This design was chosen because it enables the model to use the specifics of the before and after imagery to learn what degree of damage they indicate.

The paper starts with a description of the state of the art of automated damage assessments (Section 2). Section 3 describes the methodology and the set up of our experiments. Our experiments include 175,289 buildings, a subset of the unique xBD satellite dataset [14], that are human-labeled to four damage classes, as described in Section 4. The damage occured across three continents and is due to 13 disasters and five natural hazard types. Our first set of experiments assesses the performance across the 13 disasters (Section 5.1). These experiments include multiclass versus binary training, an inspection of what causes the differences in performance between disaster (types), and an assessment of different ways to combat class imbalance. The second set of experiments (Section 5.2) studies the transferability of learned weights to other disaster types that are not included in the training set. Sections 6 and 7 provide the discussion and conclusions and future work, respectively.

## 2. Previous Research

Automated building damage classification has received increased attention during the last several years both from practitioners [15–17] and from researchers [18,19]. Models have been developed that forecast the building damage before the disaster arrives or that predict the building damage

after the disaster has struck. Hereby, the term prediction refers to estimating the output for unseen input data with the same temporal dimensions of the input and output data, and forecasting refers to whether the output represents a state later in time than the input does [20]. Models make use of either remote sensing or geospatial data, or combinations thereof. Geospatial data are defined as data with explicit geographical locations. Additionally, remote sensing imagery contains geospatial data in terms of the bounding box of the images and the spatial reference system; however, in this case no explicit locations or objects are identified beforehand [21]. Most commonly, remotely sensed data are used to classify damage, but also other sources of geospatial data can be used as predictors, such as information on building structures [22–25]. Harirchian, Lahmer and Rasulzade [24] developed an artificial neural network (ANN) that uses buildings' damage-inducing parameters, such as number of floors, to predict the actual observed damage. The objective of the ANN model is not to be used in operational conditions after an earthquakes strikes but to identify earthquake-susceptible buildings for disaster risk management programs. Machine learning techniques, such as multiple linear regression or ANN, have also been used by, for example, the Red Cross to both forecast and predict building damage to houses due to typhoons in the Philippines [26]. The forecasts are used to trigger early action and the predictions are used to target humanitarian response. The ca. 40 predictors use geospatial data, including hazard (typhoon wind speed, track, and rainfall), exposure (population density, number of households), topography and geomorphology (slope, ruggedness, elevation) and vulnerability features (roof material, wall material). This approach is fast, but applicable only for one single hazard type and coarse, as it only allows to predict damages at an aggregated subprovincial level. Gunusekera et al. [27] classified the different techniques available to predict damage along an axis of speed of delivery and level of detail.

The focus of our research is on a technique that has a high speed of delivery and considerable level of detail. In addition, we focus on a technique with minimum data requirements and multi-hazard and spatial transferability. This requires a focus on remote sensing data. As remotely sensed data, different image sources with different resolutions can be used, where most studies focus on imagery obtained from satellites and UAVs. Many of the techniques for automated damage classification can be applied to both source types [13] and the combination can improve the performance of this task [28]. Our research solely focuses on imagery from satellites, because these are more widely available and can be obtained directly after a disaster, and a larger set of labeled satellite data exists.

The first studies on automatically predicting damage from remotely sensed data mainly explored the use of texture- and segmentation-based methods (e.g., [23,29–31]) and showed that while it is possible to automatically detect damage, results in performance are not always satisfying [18]. Due to the complex nature of features indicating damage and with the increased popularity of machine learning, research started to be conducted on the use of machine learning techniques for automatic classification, especially focusing on CNNs. Though relatively little comparison work has been done, CNN-based models were shown to outperform texture- and segmentation-based methods [32,33] and other machine learning approaches [34–36], including non-CNN-based neural networks [37,38]. Research on CNN-based models for damage classification has focused on improving speed and usability [39], alleviating the effects of imprecise building masks [40], and extending the classification to man-made building features connected to damage, such as blue tarp cover [41]. While the use of CNNs has shown the most promising results in previous research, no models using these novel technologies have yet been applied in real-time operations [18].

Table 1 shows an overview of the research done on automated damage classification with satellite imagery using CNNs on the spatial granularity scale of buildings. For each work, the best performing model when trained and tested on the same disaster is shown. The performance is measured as accuracy since this is the only metric reported across all articles. Unfortunately, accuracy is not a very good performance metric for this problem, because of the high class imbalance: in most disasters, undamaged buildings are much more frequent than damaged ones. Nonetheless, it can be concluded that all models are learning damage-related features, since their accuracy is larger than that of a naive

classifier that always predicts the majority class. At the same time, the quality of the different models cannot be directly compared based on the reported accuracy, since there are more differences between the studies than solely the model; these include class balance, disaster type, and data sources. We also note that over the years models have become more complex [13,32,42–45], but this has not led to an evident increase in accuracy, as shown in Table 1.

All studies summarized in Table 1 solely discriminated two classes, making a distinction between *destroyed* and *not destroyed* buildings. The reason given for this is that other levels of damage are hard to distinguish from satellite imagery and that the definitions of more than two classes can become very ambiguous [18,46,47]. It is important to verify whether this is true. The damage classes should fit the purpose of the assessment and while binary labels can in some cases be enough, it is highly likely that a more precise distinction is needed for other purposes.

As a source for the damage labels, most studies used the analysis done by UNITAR/UNOSAT [32,44,45]. These data have been created through the annotation of satellite imagery by professional analysts and are openly available through the Humanitarian Data Exchange (HDX) (https://data.humdata.org). The original UNOSAT assessments used in the cited studies differentiate five damage levels, but all cited studies binarize those five levels to two. Three out of four studies that use the UNOSAT data grouped all classes other than *destroyed* to one class, while the authors of [44] also included the level of *severe damage* in the damaged class. The class consisting of *destroyed* and possibly *severe damage* buildings is from here on referred to as *destroyed* and the other class as *not destroyed*. When inspecting the sources of satellite data, all studies except [42] used data from the Maxar/DigitalGlobe Open Data Program. The ground sample distance (GSD) differs per Maxar satellite but mostly varies between 0.3 and 0.7 m. As can also be seen in Table 1, all studies solely used information from the visible spectrum, namely the RGB bands, and four out of six used imagery before and after the disaster in their model, whereas two solely used after imagery.

Most studies worked with an imbalanced dataset with respect to the difference between the number of *destroyed* and *not destroyed* buildings [32,42,44,45], as shown in Table 1. The two studies that had a balanced dataset [13,43] prepared this on purpose by selecting solely a part of the original data, and the same was done, to a certain extent, by the other studies to diminish the imbalance. Such a selection is not possible during a real-time emergency response, and thus it is likely that in such a scenario the class imbalance is large. The effect of class imbalance is hard to assess from previous research. It could be argued that a larger imbalance improves performance when measured in terms of accuracy since the model tends to lean towards the majority class. To know if the model indeed shows this behavior, the performance of all classes should be analyzed separately. Only two studies reported this, and in both cases the model performed better on the majority class (*not destroyed*) than on the minority (*destroyed*) class [32,45], suggesting that a larger class imbalance can lead to better accuracy while not indicating a better predictive value.

Regarding the type of natural hazard, all studies solely examined one type, and thus no model has been tested across different types of natural hazards. Combined they covered three types: *earthquake, tsunami,* and *hurricane*, where *earthquake* is the most common type. Among all studies shown in Table 1, those focusing on earthquakes [13,32,37,44,45] reported the lowest performance. However, since all models have only been tested on one hazard type, it should be further researched if and how the disaster type influences the performance.

**Table 1.** Overview of research done on automated building damage assessment from satellite imagery using neural networks. *Maxar* refers to the Maxar/DigitalGlobe Open Data Program. All publications predict damage as a binary distinction, namely *destroyed (destr.)* vs. *not destroyed (not)*.

| Reference and Publication Date | Accu-Racy | Model | Satellite Data | | | | Labeled Data | | Label perc. (destr./not) |
| | | | Disaster Type | Source | Time | Bands | Source | Type | |
|---|---|---|---|---|---|---|---|---|---|
| [42] 05-2017 | 95.6 | Siamese CNN (4 layers) | tsunami | PASCO | pre & post | RGB | government | field survey | 40/60 |
| [43] 05-2019 | 97.3 | CNN (4 layers) | hurricane | Maxar | post | RGB | Tomnod | remote sensed | 50/50 |
| [32] 05-2019 | 87.6 | CNN (3 layers) + Random Forest | earthquake | Maxar | pre & post | RGB | UNITAR/ UNOSAT | remote sensed | 36/64 |
| [44] 10-2019 | 78.0 | Siamese CNN plus CNN after concatenation | earthquake | Maxar | pre & post | RGB | UNITAR/ UNOSAT | remote sensed | 55/45 |
| [13] 11-2019 | 87.1 | CNN (adaptation of densenet121) | earthquake | Maxar | post | RGB | authors | remote sensed | 50/50 |
| [45] 01-2020 | 88.8 | CNN (VGG16) | earthquake | Maxar | pre & post | RGB | UNITAR/ UNOSAT | remote sensed | 34/66 |

All results shown in Table 1 have been trained and tested on the same disaster, but Nex et al. [13] and Xu et al. [44] also experimented with training on a set of disasters and testing on another disaster. Both studies focused on earthquake damage but used a mix of disasters with different geographical locations. Both papers found that the performance dropped when not including the test disaster in the training set. Nex et al. [13] attributed this difference to differences in image quality and building topology, but did not quantify it. Both works also found that the ability to transfer the trained model to another disaster heavily depends on the combination of the training and testing disaster(s). For the best combination of training and testing disasters, the model used by Xu et al. [44] reached 92% of the accuracy that was obtained when the model was trained on the test disaster itself.

Both studies also applied fine-tuning as an experiment, meaning that they first train the model on a set of disasters and thereafter do the final training on a limited set of the test disaster. Xu et al. [44] used 10% of the available buildings for the fine-tuning, whereas Nex et al. [13] used between 15% and 25%. Both studies showed that fine-tuning improves performance in all setups. However, in most emergency response contexts having this percentage of labeled data of the test disaster is often not realistic due to time and resource constraints.

## 3. The CNN-Based Model

The model designed for this research takes into account the limited data availability and builds on previous research. Regarding the data availability, RGB satellite imagery is the most commonly available imagery source, and therefore the model architecture is optimized for this data source. As discussed in Section 2, a CNN model was shown to gain the best performance for damage classification. Moreover, CNNs have a relatively good transferability compared to other machine learning approaches [48]. The proposed model uses before and after imagery as input to the model, since previous research showed that this gives good results and we hypothesized that including both snapshots improves the transferability of the model.

A schematic overview of the designed model architecture is shown in Figure 2. The model takes as input the pixels of a building before and after the disaster, where the image consists of the RGB bands. The two images are given as input to two separate CNNs, whereafter the output feature vectors are concatenated. The concatenated feature vector is then given as input to two fully connected blocks, each consisting of a fully connected layer, a rectified linear unit (ReLU) activation function [49], batch normalization [50], and dropout [51]. The last layer is yet another fully connected layer, followed by a Softmax activation to produce the output. This output is a damage label, where the number of damage classes depends on the distinction made in the train data. In this research, the number of damage classes is either two or four, depending on the experiment. The architecture is inspired by a siamese network [52,53], but differs from such a network in two aspects. Firstly, the two CNNs have separate weights, whereas in a siamese network they share the weights. This choice was made because we believe the before and after imagery have distinct features that require separate weights to be extracted well, and this was also shown in preliminary experiments. Moreover, in a siamese network the final outcome is solely based on the differences in the output vectors of the CNNs, but in our design it was chosen to let the model learn the distance function by adding three fully connected layers. This enables the model to learn which features from each CNN indicate different damage classes.

To generate the input images, it is assumed that the building polygons on the before imagery are known. Based on these building polygons, individual building images are generated by selecting a bounding box around the polygon and then adding 20% to the width and height of the bounding box. This extra 20% around the building is taken because the surrounding area often contains important information about the damage, such as debris and surrounding water, and because the building polygons do not always precisely overlap with the buildings in the post imagery. While this extra area around the building can also lead to adverse information, for example by showing washed away debris from other buildings [54], we chose to expand the bounding box as it has been shown in previous research that it does increase the performance for object classification from satellite imagery [55].

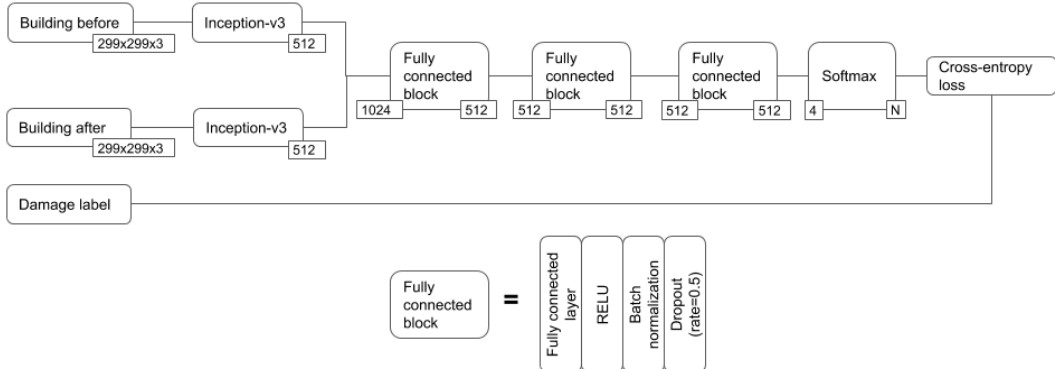

**Figure 2.** Architecture of the used model. The numbers in the squares indicate the input and output size of each block, where N equals the number of damage classes.

The *before* and *after* building images are given as input to two separate CNNs, which follow the Inception-v3 architecture [56]. Inception was chosen as the CNN architecture since it has a relatively low computational cost and works with different kernel sizes such that it can detect features of different sizes, and works well with inputs that represent different scales [56]. The input size of the Inception model is 299 × 299 pixels and since the whole architecture has to be modified if reducing the input size, it was chosen to re-scale all cropped building images to have a width and height of 299 pixels. The output feature vector of Inception-v3 has a size of 2048, but in our model this output vector was scaled down to have a size of 512. This lowering of the output dimension was done to prevent overfitting, since most original input images are smaller than 299 × 299 pixels and thus most information can be retrieved with a 512 feature vector. The weights of the Inception-v3 models were pre-trained on ImageNet [57], since this has been shown to often improve the accuracy drastically [58,59]. The weights are re-trained on the disaster data during the training process.

Several configurations related to model hyperparameters and data augmentation techniques were tested in our preliminary studies and the best performing one was chosen for conducting the experiments in this study. The used configuration includes a simple data augmentation scheme containing rotation, flipping, and translation. The optimizer used was Adam, with the suggested hyperparameter values [60]; the batch size was 32 and the model was trained for 100 epochs. All code and its settings are openly available (https://github.com/rodekruis/caladrius) in Supplementary Materials. All experiments were run on a single GPU, most of them using the Nvidia V100 Tesla card with 32 GB memory.

In this research, we differentiate between multiclass and binary classification. For assessing the results of the multiclass classification, the macro F1 [61], harmonic F1, and recall per class are used as performance measures. Macro F1 was chosen because it is the most common measure for the multiclass setting and represents the overall performance well. The harmonic F1 gives more importance to equal performance across all classes, which is relevant in our context if we want to make sure none of the classes are very wrongly classified. Moreover, the recall per class is looked at, because to assess the suitability of the model for practical purposes it is important to know to what extent the model is recognizing the damage in each class. For the binary classification setting, area under curve (AUC) and accuracy are used to assess the performance. Accuracy is the most commonly reported performance measure in other studies, although it has to be interpreted with care given the high imbalance of classes. Hence, AUC is included since it is less sensitive to imbalance.

## 4. Data

The data used in this research are part of the xBD dataset [14], covering 19 disasters and 850,736 buildings. Te data consist of optical satellite images with RGB bands and a resolution equal to or less than 0.8 m GSD. All images employed were sourced by the Maxar/DigitalGlobe Open Data Program (https://www.digitalglobe.com/ecosystem/open-data). Moreover, the xBD dataset consists

of building polygons, each having a damage label. These polygons and labels were produced manually, see [14] for more information on the labeling process. We chose to focus on five of the seven natural hazards types that are present in the xBD dataset, namely *tornadoes, hurricanes, floodings, tsunamis,* and *volcanic eruptions*, because these are in line with the priorities of the Red Cross, also in terms of predictive impact modeling. This left us with 13 disasters, key information of which is displayed in Table 2. The *Disaster* column indicates the name and natural hazard type of the disaster. The five different types of natural hazards the studied disasters are caused by can be divided into four different damage types, namely *flooding, wind, tsunami,* and *volcano*, as indicated in the *Damage type* column in Table 2. Each disaster belongs to one hazard type and one damage type. The 13 disasters are spread over three regions—North America, Central America, and Asia. Only the data in the first release of the xBD dataset were used, unclassified buildings were removed, and buildings that had more than 10% zero-valued, i.e., black, pixels were also removed. This left us with 175,289 buildings. The buildings were labeled according to the Joint Damage Scale, which distinguishes four classes: *no damage, minor damage, major damage,* and *destroyed* [14]. This is a newly developed scale, which was specially designed to be generalized for damage caused by different types of natural hazards and to be well-suited for annotation from aerial imagery. Commonly used scales, such as the EMS-98 [62], only cover one type of damage and categorize according to damage that is not always visible from the sky [46,63]. Besides the location of the buildings and their degree of damage, the xBD dataset does not include other information on characteristics such as the type of buildings. In the used dataset, the percentage of buildings belonging to each class is highly imbalanced, where for most disasters the majority of buildings belong to the *no damage* class, as can be seen in Table 2.

**Table 2.** Details of the disasters included in this research. The column *Class distribution* shows the percentage of samples belonging to the classes *no damage, minor damage, major damage* and *destroyed*.

| Region | Disaster | Damage Type | Number of Buildings | Class Distribution |
|---|---|---|---|---|
| Asia | | | | |
| | Nepal flooding | flooding | 29,808 | 75/13/11/1 |
| | Palu tsunami | tsunami | 24,119 | 83/0/2/15 |
| | Sunda tsunami | tsunami | 11,682 | 98/0/1/1 |
| Central America | | | | |
| | Guatemala volcano | volcano | 493 | 94/1/1/4 |
| | Hurricane Matthew | wind | 9506 | 19/52/16/12 |
| North America | | | | |
| | Moore tornado | wind | 18,858 | 87/4/2/7 |
| | Tuscaloosa tornado | wind | 12,579 | 75/14/3/8 |
| | Joplin tornado | wind | 12,165 | 56/15/7/22 |
| | Hurricane Michael | wind | 20,046 | 64/24/9/3 |
| | Hurricane Florence | flooding | 5243 | 77/2/20/1 |
| | Hurricane Harvey | flooding | 21,516 | 50/12/36/2 |
| | Midwest flooding | flooding | 7161 | 96/2/1/1 |
| | Lower-Puna volcano | volcano | 2113 | 81/2/1/16 |
| 3 | 13 | 4 | 175,289 | 72/12/10/6 |

While the xBD dataset is of very high quality, there will always be features of the data that are not optimal and that the model has to learn to manage. Some of these challenges are shown in Figure 3. Firstly, environmental effects such as clouds covering buildings, as shown in Figure 3a, can be challenging. Although we did not exclude images partially covered by clouds from the analysis, we verified by manual inspection that these constitute only a very small fraction of the total dataset. Secondly, the building outlines labeled on the pre-disaster imagery and the post-disaster images can be shifted, mainly due to differences in the off-nadir angle of the satellite. This can cause the building shape polygon not to overlap with the actual building, as shown in Figure 3b, and thus, in this case, only partial information is given to the model. Thirdly, the manual labeling of buildings, in this case

done by humans from aerial imagery, remains a very subjective task. Whether a building belongs to one or another class is debatable, but clear mistakes in the labels can be seen for some buildings where the building is split into two polygons that have been assigned different labels, see Figure 3c. Though this is an exception, it indicates that the model also has to deal with these kinds of ambiguities in the data. Lastly, the differences in satellite parameters (resolutions, angles, and elevations) can cause images to look very different (an interested reader can find more information on remote sensing geometry in, e.g., [64]). The used dataset has a large variety in these parameters, as is displayed in Table 3. Figure 3d shows an example of the effects of the large differences between datapoints in this example in terms of the level of illumination.

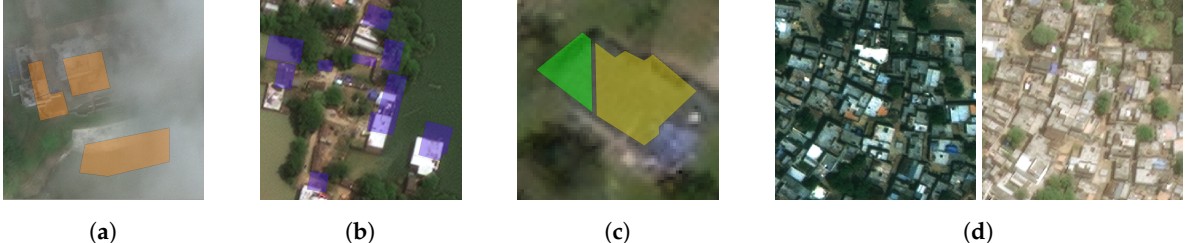

|      (a)       |      (b)       |      (c)       |      (d)       |

**Figure 3.** Examples of challenges in the data. The type of challenge is indicated by the subcaption. (**a**) Cloud cover; (**b**) misaligned building polygons; (**c**) building with two different damage labels; (**d**) difference in illumination on before (**left**) and after (**right**) images.

**Table 3.** The median, mean, and standard deviation of several important image parameters across the studied data.

| Parameter | Median | Mean | Standard Deviation |
|---|---|---|---|
| Image pairs per disaster | 2.00 | 2.38 | 1.50 |
| Disaster duration (days) | 16.00 | 51.29 | 56.49 |
| Panchromatic resolution (m) | 0.51 | 0.52 | 0.12 |
| Off-nadir angle (degrees) | 21.75 | 22.92 | 10.28 |
| Sun elevation (degrees) | 62.53 | 60.82 | 11.04 |
| Target azimuth (degrees) | 177.35 | 173.43 | 106.68 |
| Sun azimuth (degrees) | 140.52 | 132.22 | 31.69 |
| Building footprint (m$^2$) | 207.90 | 297.63 | 752.33 |
| Delay pre and post imagery (days) | 351.00 | 561.71 | 515.37 |
| Delay start disaster and post imagery (days) | 12.00 | 31.6 | 48.69 |

Figure 4 shows the before imagery, after imagery, and damage labels of a small area impacted by the Joplin tornado and the Nepal flooding, as an example. It can be seen that both areas were heavily damaged, while the damage looks completely different. In addition, we see a clear difference in the distribution of damage and the spatial ordering of the buildings. We can examine the appearance of the buildings and damage further by inspecting individual buildings. Figure 5 shows an example of a building in each damage class after the disasters for the Joplin tornado and the Nepal flooding. For the Joplin tornado, it is visible that as the damage level increases, the damaged buildings appear browner, indicating broken roofs. In the *minor damage* class, a characteristic sign is the blue tarps on the roof. For the Nepal flooding, the color distribution is different: as the damage class increases, the images appear more green, which indicates the presence of water and thus (partly) flooded buildings. Due to the relatively high off-nadir angle and the fact that the building polygons were marked on the before imagery, not all buildings are well-captured by the building bounding box, such as in Figure 5f. This is the nature of the data and an extra difficulty for the model. Moreover, some images are less sharp for the Nepal flooding than the Joplin tornado, which is due to the higher off-nadir angle and the often smaller building sizes.

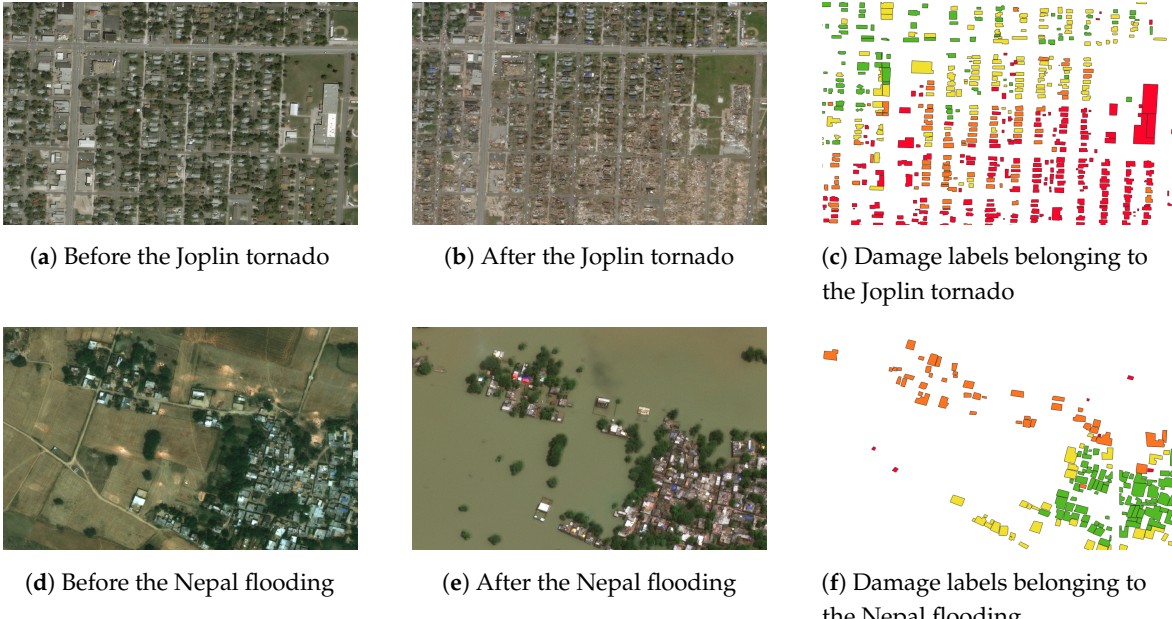

(**a**) Before the Joplin tornado    (**b**) After the Joplin tornado    (**c**) Damage labels belonging to the Joplin tornado

(**d**) Before the Nepal flooding    (**e**) After the Nepal flooding    (**f**) Damage labels belonging to the Nepal flooding

**Figure 4.** Before and after imagery and damage labels of part of the areas that were impacted by the Joplin tornado (**top row**) and the Nepal flooding (**bottom row**). The colors of the damage labels in figures *c* and *f* indicate the different degrees of damage, where green equals *no damage*, yellow *minor damage*, orange *major damage*, and red *destroyed*.

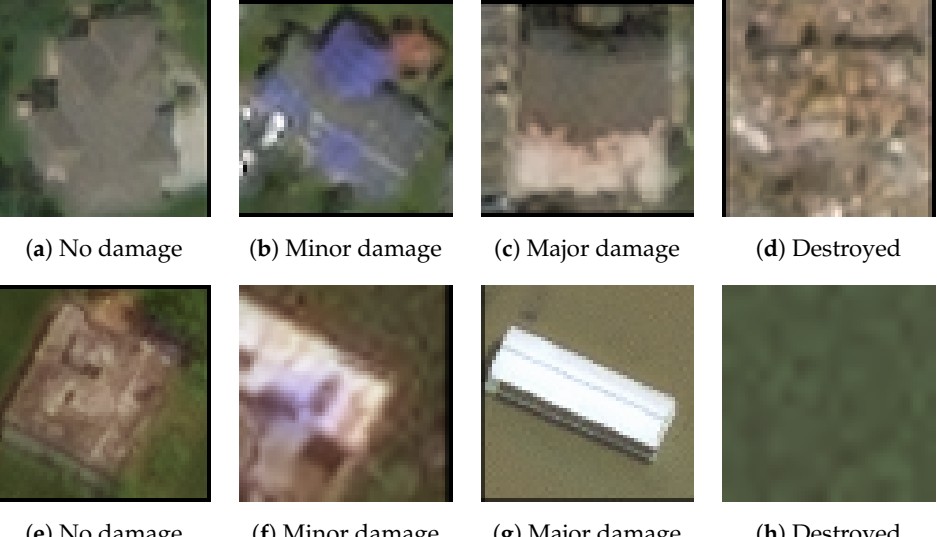

(**a**) No damage    (**b**) Minor damage    (**c**) Major damage    (**d**) Destroyed

(**e**) No damage    (**f**) Minor damage    (**g**) Major damage    (**h**) Destroyed

**Figure 5.** Examples of individual buildings after the disaster impacted by the Joplin tornado (**a–d**) and the Nepal flooding (**e–h**). The subcaptions indicate the damage class.

## 5. Experiments and Results

The experiments of this work can be divided into two parts, namely testing the performance of the model (i) on a variety of disaster data, as described in Section 5.1, and (ii) when excluding the test disaster from the training set, as described in Section 5.2.

The goal of (i) was to understand how the model performs on different disasters, and if differences in performance can be explained by certain data parameters. To explore this notion, for each of the 13 disasters the model was trained on 80% of the data and tested on 10%; the remaining 10% were used for validation, i.e., for tuning model hyperparameters, such as the number of training epochs. This was done for both multiclass and binary classification. The 13 disasters differed from each other on three aspects: *damage type, geographical location*, and *image parameters*. For (ii), the goal was to see

how well a model trained on a set of disasters performs on a test disaster that the model had not been trained on. Two of the 13 disasters were taken as test disasters. For both test disasters, models trained on different sets of disasters were assessed.

### 5.1. Performance on 13 Different Disasters

Table 4 shows the results of training and testing the model on all 13 disasters separately. The training time ranged between 20 min and 10 h, depending on the training data size. Producing the test results took between three and 26 s. From Table 4, large differences in performance can be seen between disasters. For four disasters, the harmonic, as well as macro F1, were larger than 0.7. On the other hand, the harmonic F1 score of five other disasters was zero, caused by the fact that the recall of at least one of the classes was zero. The recall per class gives insightful information in assessing difficulties, for example indicating that the model often gains a higher recall for the majority class. This dependence is further discussed in Section 5.1.2 and thereafter experiments to tackle the negative effects of the imbalance are executed in Section 5.1.3.

**Table 4.** Results of multiclass classification per disaster, sorted by the harmonic F1 score. *No, Min., Maj.,* and *Des.* indicate the *no damage, minor damage, major damage,* and *destroyed* classes, respectively.

| | F1 | | | Recall | | | | Class Percentage | | | |
|---|---|---|---|---|---|---|---|---|---|---|---|
| **Disaster** | **Harm.** | **Macro** | **No** | **Min.** | **Maj.** | **Des.** | **No** | **Min.** | **Maj.** | **Des.** |
| Sunda | 0.000 | 0.332 | 0.997 | 1.000 | 0.000 | 0.286 | 98.7 | 0.0 | 0.7 | 0.6 |
| Midwest | 0.000 | 0.372 | 0.999 | 0.000 | 0.429 | 0.000 | 97.0 | 1.7 | 1.0 | 0.3 |
| Puna | 0.000 | 0.449 | 0.980 | 0.000 | 0.000 | 0.800 | 78.6 | 1.6 | 1.6 | 18.2 |
| Florence | 0.000 | 0.466 | 0.975 | 0.000 | 0.922 | 0.000 | 76.8 | 2.3 | 19.9 | 1.0 |
| Palu | 0.000 | 0.702 | 0.981 | 1.000 | 0.000 | 0.824 | 83.8 | 0.0 | 1.8 | 14.4 |
| Nepal | 0.073 | 0.482 | 0.989 | 0.010 | 0.459 | 0.368 | 74.3 | 13.0 | 12.1 | 0.6 |
| Harvey | 0.381 | 0.538 | 0.899 | 0.162 | 0.861 | 0.143 | 50.5 | 12.1 | 36.1 | 1.4 |
| Michael | 0.495 | 0.542 | 0.914 | 0.401 | 0.413 | 0.352 | 65.0 | 24.0 | 8.4 | 2.6 |
| Matthew | 0.543 | 0.577 | 0.382 | 0.858 | 0.315 | 0.676 | 16.9 | 50.0 | 18.1 | 15.0 |
| Tuscaloosa | 0.706 | 0.740 | 0.952 | 0.716 | 0.500 | 0.781 | 74.4 | 16.1 | 3.7 | 5.8 |
| Joplin | 0.758 | 0.787 | 0.961 | 0.705 | 0.516 | 0.907 | 55.9 | 15.6 | 7.4 | 21.1 |
| Moore | 0.771 | 0.802 | 0.995 | 0.500 | 0.766 | 0.887 | 87.1 | 4.0 | 2.4 | 6.5 |
| Guatemala | 1.000 | 1.000 | 1.000 | 1.000 | 1.000 | 1.000 | 93.8 | 0.0 | 0.0 | 6.2 |

To examine the results further, we can look at the confusion matrices. Figure 6 shows the confusion matrices for two disasters, the Nepal flooding and the Joplin tornado. In the confusion matrix of Nepal, we can see that the model had a high tendency to predict *no damage*, even for *destroyed* samples. This is non-desirable behavior as the model did not learn the correct damage-related features. To understand where the mispredictions came from, a qualitative analysis can be done. Figure 7 shows four examples of misclassified buildings; the top row shows two buildings where the damage is under-predicted, while in the bottom row the damage is over-predicted. While the model should correctly classify these examples, it can also be seen that some buildings are challenging for determining their damage level. For example, the building in Figure 7a does not appear to be fully surrounded by water due to the trees, judging from the area of the building crop, leading to the predicted label of *no damage*. However, if zooming out, it could be seen that the building is fully surrounded by water, leading to a *major damage* ground-truth label. The building in Figure 7c is blurred due to the small building size, which makes it hard to determine the level of damage. Given the limited information, the damage could be interpreted as *destroyed* instead of *no damage*, as the model did, since the post image does appear completely flooded such that no building is visible anymore, and the white pixels on the pre-image might be interpreted as the original building. The model overpredicted damage in Figure 7d, possibly due to the lighting that makes the field appear like water. Nevertheless, the model is also making mistakes that should be recognizable, such as the building in Figure 7b. This qualitative analysis

indicates that while the model did not learn all relevant features, not all data points with damage showed clear damage-related features.

For Joplin, the confusion matrix showed a different pattern than for Nepal. In general, more samples were classified correctly and if they were misclassified, they were often classified as one of the adjacent classes. This is a more desirable behavior and indicates that the model learned relevant features from the data. Again, examining a few samples in Figure 8 helps to understand the misclassifications. Sometimes the model even outperforms the human annotators, such as in Figure 8c where the ground truth label was clearly wrong. The building in Figure 8a shows minor damage, but this is difficult to differentiate due to not showing big holes or a blue tarp, which is the case for most minor damage caused by the Joplin tornado. The building in Figure 8d was wrongly predicted as destroyed, and this is probably due to the debris in front of the building. The building in Figure 8b was clearly mispredicted and it is ambiguous why the model mistook this building as showing no damage. One possible explanation could be that most damaged buildings show brownish damage features, which is not the case here. However, if this is the explanation it indicates that the learned representation of damage features was not fully correct. Concluding, the model seemed to learn most damage-related features of the Joplin tornado well, sometimes even outperforming the human annotators.

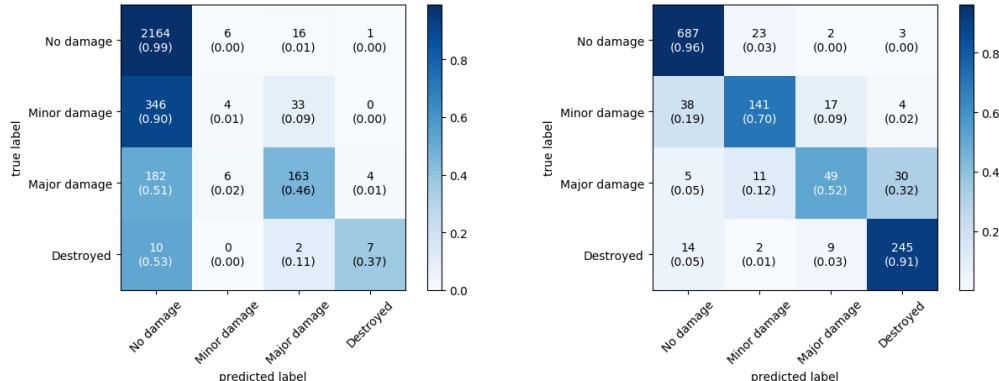

**Figure 6.** Confusion matrices of the model tested on the Nepal flooding (**left**) and the Joplin tornado (**right**). Both models were trained on 80% of the data, and tested on 10%.

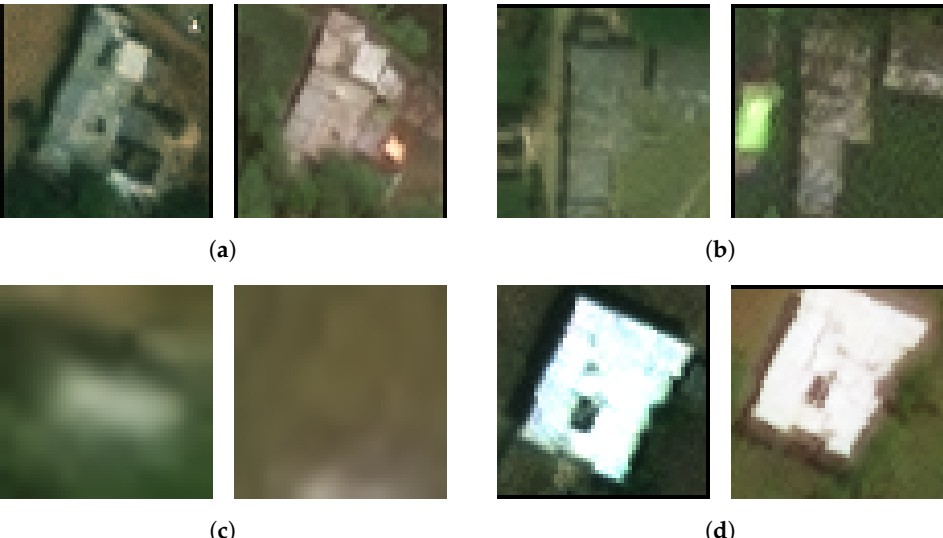

**Figure 7.** Four examples of buildings in the test set of the Nepal flooding that were misclassified by the model. For each building, the before image is shown on the left and the after image on the right. (**a**) Label: *major damage*; prediction: *no damage*; (**b**) Label: *major damage*; prediction: *no damage*; (**c**) Label: *no damage*; prediction: *destroyed*; (**d**) Label: *no damage*; prediction: *major damage*.

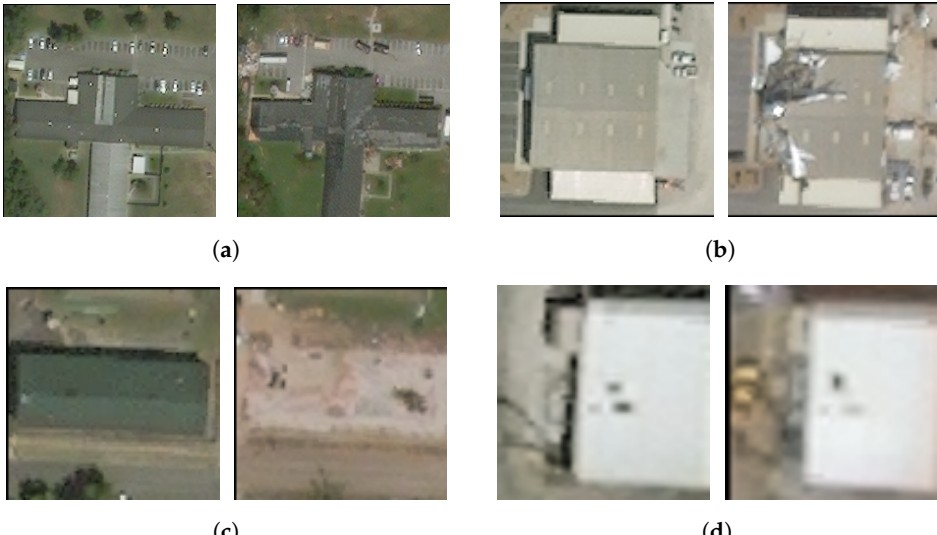

**Figure 8.** Four examples of buildings in the test set of the Joplin tornado that were misclassified by the model. For each building, the before image is shown on the left and the after image on the right. (**a**) Label: *minor damage*; prediction: *no damage*; (**b**) Label: *major damage*; prediction: *no damage*; (**c**) Label: *no damage*; prediction: *destroyed*; (**d**) Label: *no damage*; prediction: *destroyed*.

### 5.1.1. Multiclass versus Binary Training

Since in the previous studies the focus was on the binary distinction between *not destroyed* vs. *destroyed*, and as this might suffice in operational use for a subset of purposes, we also tested the performance of the model on all 13 disasters for this binary distinction. The performance for binary labels was calculated in two ways. Firstly, the model was trained on binary labels, and secondly, the model was trained on multiclass labels and only after the predictions the labels were grouped into binary labels. In this case, the *no damage, minor damage*, and *major damage* classes were combined to the *not destroyed* class.

The results for both experiments are displayed in Table 5. This table shows that a higher or similar AUC was gained when training on the multiclass labels as compared to binary labels, while the differences in accuracy were small. With accuracy for the binary classification, the building was assigned to the *destroyed* class if the probability calculated by the model of belonging to that class was larger than 0.5. With AUC this class threshold was varied, and thus the relatively small differences in accuracy compared to the differences in AUC indicate that the model can separate the classes better at another threshold than that of 0.5. Training the model on multiclass labels and thereafter grouping them to a binary distinction was the procedure applied in the remainder of experiments where binary classification was done.

**Table 5.** Binary AUC and accuracy scores. *Multiclass* refers to the model trained on multiclass labels but grouped into binary labels for performance measures. *Binary* refers to the model trained on binary labels. The numbers are sorted by the difference in AUC of the binary model minus the AUC of the multiclass model.

| | AUC | | Accuracy | |
|---|---|---|---|---|
| **Disaster** | **Multiclass** | **Binary** | **Multiclass** | **Binary** |
| Florence | 0.874 | 0.496 | 0.988 | 0.990 |
| Sunda | 0.869 | 0.694 | 0.993 | 0.994 |
| Harvey | 0.897 | 0.853 | 0.987 | 0.987 |
| Matthew | 0.936 | 0.894 | 0.904 | 0.895 |
| Michael | 0.926 | 0.891 | 0.974 | 0.976 |
| Nepal | 0.920 | 0.905 | 0.994 | 0.990 |
| Puna | 0.975 | 0.962 | 0.948 | 0.943 |
| Moore | 0.997 | 0.990 | 0.989 | 0.986 |
| Joplin | 0.990 | 0.987 | 0.952 | 0.941 |
| Guatemala | 1.000 | 1.000 | 1.000 | 1.000 |
| Tuscaloosa | 0.990 | 0.990 | 0.971 | 0.971 |
| Palu | 0.971 | 0.972 | 0.955 | 0.957 |
| Midwest | 0.987 | 0.997 | 0.997 | 0.997 |

### 5.1.2. Understanding Differences between Disasters

The performance between disasters differed and can be evaluated for each of the performance measures, regardless of binary or multiclass labels. Trying to understand where those differences come from is important for practical purposes.

The first factor that could influence the performance is the difference in class distributions. Figure 9 displays the percentage of data points belonging to a class vs. the recall of that class for each disaster and each of the four classes. There was a monotonically increasing, roughly logarithmic, relation between the class distribution and the recall, revealing that the recall generally increased when a larger percentage of data points belonged to that class. Concurrently, not all data points followed this trend, especially in the major damage and destroyed classes.

A second factor could be the building footprint, assuming buildings with smaller footprints are more difficult to classify due to fewer optical clues. Figure 10 shows the distributions over the building footprint for the correctly and incorrectly classified buildings, for the 95% percentile of buildings. The distributions of the building footprint over correctly and incorrectly classified buildings largely overlapped. This shows that a larger building size is not a predictor for the chance that a building will be correctly classified.

The third factor could be disaster-specific parameters, such as the number of satellite image pairs covering a disaster, the number of buildings, the type of disaster, and the geographical region where the disaster struck. A technique to understand the relation between these parameters and the performance is by making a scatter plot of the value of the parameter versus the performance metric. The AUC over the binarized labels *destroyed* and *not destroyed* was chosen as a performance metric, since this is the measure least influenced by class imbalance. Figure 11 shows the scatter plots for the four parameters. From these plots, it can be seen that there was no correlation between the AUC and any of the four parameters.

Lastly, we can inspect parameters that are specific for a pre- and post-disaster satellite image pairs. These include the off-nadir angle, panchromatic resolution, sun azimuth, sun elevation, and target azimuth. In Figure 12, scatterplots for these parameters versus the AUC are shown. For each parameter a scatter plot was made for the sum of the pre and post images of that parameter, and of the absolute difference between the pre and post images of that parameter. Again, we can see that none of the parameters had a very distinct relationship with the AUC. The largest influence seems to come from the sum of the Sun azimuth; the larger this sum, the smaller the AUC. This same relation is

somewhat visible in the difference in sun elevation. Both these parameters have to do with lighting and thus it can be interpreted that too little/too much lighting or larger differences in lighting can degrade performance.

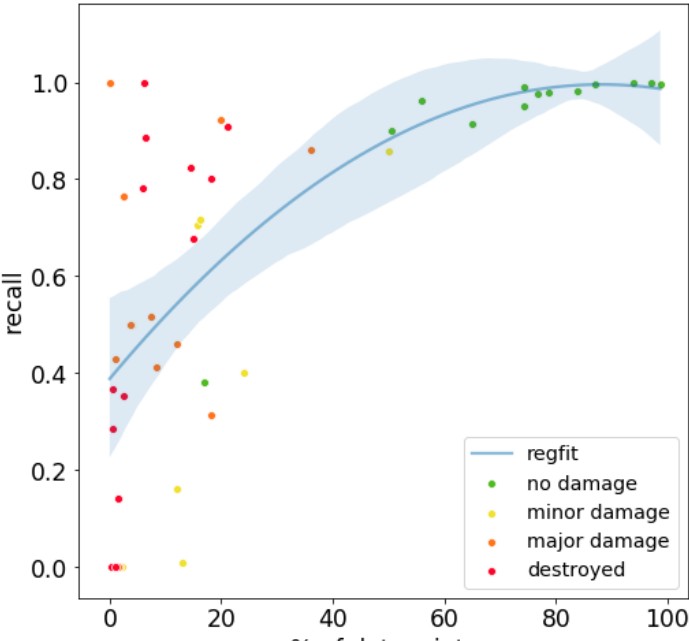

**Figure 9.** Scatter plot of the percentage of data points belonging to a class versus the recall of that class for each of the 13 tested disasters. The blue line shows the best polynomial fit and the blue area the 95% confidence interval.

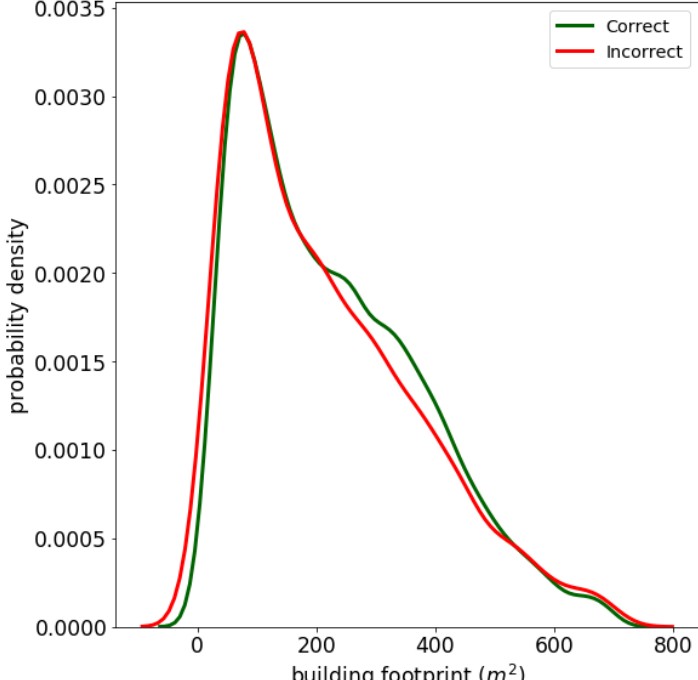

**Figure 10.** Distribution plot of the building footprint for buildings up to 700 m$^2$, i.e., 95% of the buildings. The lines correspond to the distributions over correctly and incorrectly classified samples.

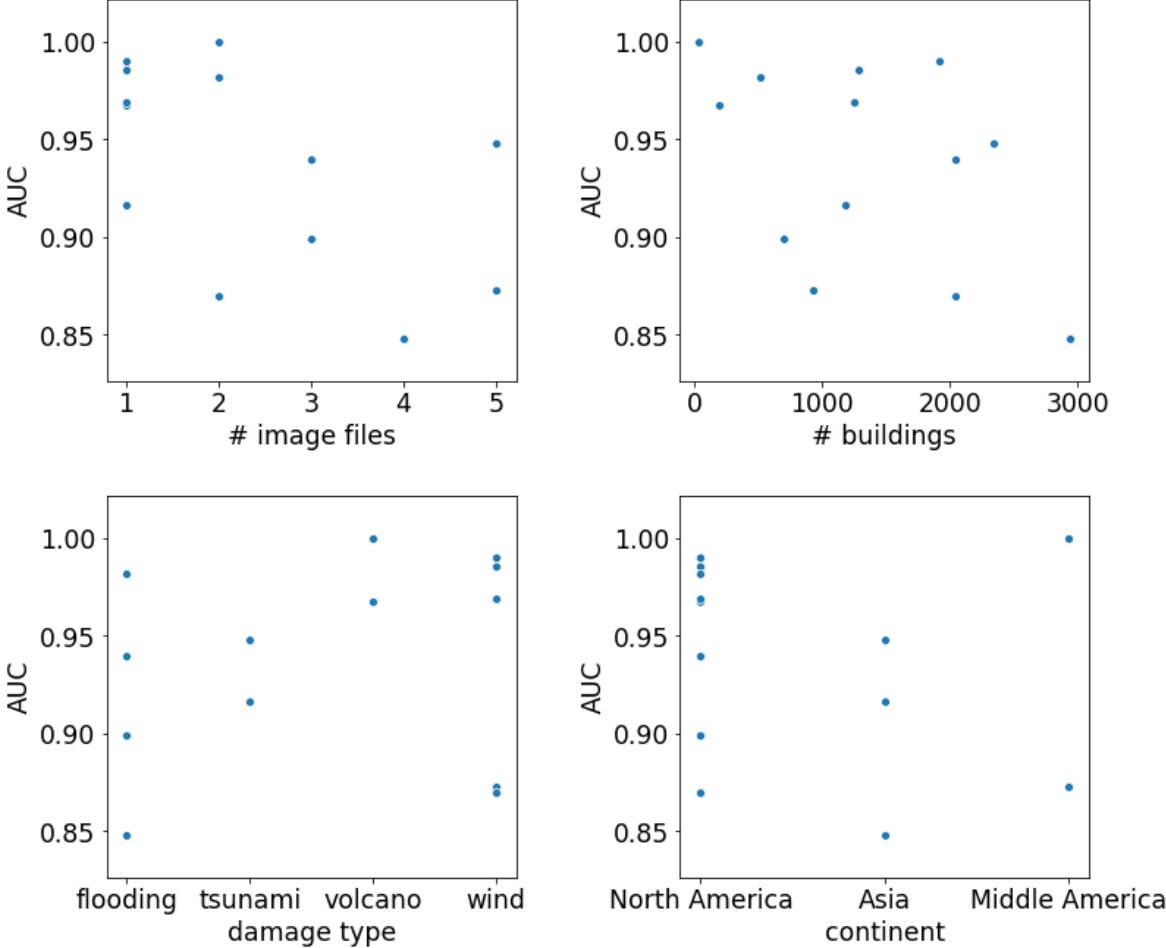

**Figure 11.** Scatter plots of the value of the parameter versus the AUC. One dot belongs to one disaster.

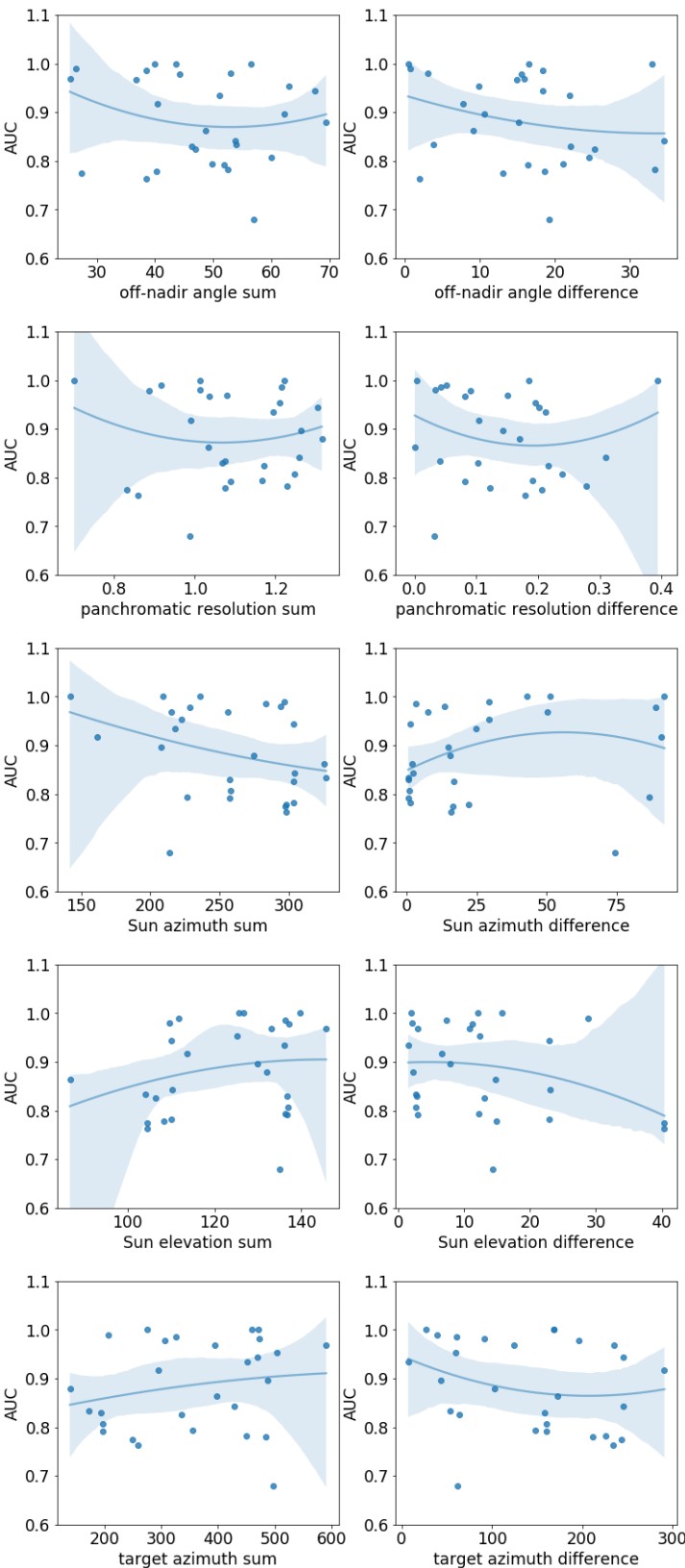

**Figure 12.** Scatters plot of the value of the parameter versus the AUC. One dot belongs to one pre-disaster–post-disaster satellite image pair. The *x*-axis of the left column represents the sum of the parameter over the pre and post images. In the right column, the *x*-axis represents the absolute difference in the parameter value between the pre and post images.

### 5.1.3. Combating Loss in Performance Due to Class Imbalance

The class imbalance had a negative influence on the recall per class, as shown in Figure 9, which is harmful in our context since often the minority classes are more important to be predicted correctly. To limit the decrease in performance due to the imbalance, two methods commonly used to combat class imbalance were implemented, namely resampling and cost-sensitive learning [65].

As a resampling method, balanced resampling was applied. This means that a combination of up- and downsampling was done such that all classes contained the same amount of samples. In this implementation, the total number of samples was left unchanged in the resampled data compared to the original data. As a second method, cost-sensitive learning was applied. With this method, the balance of samples is not changed, but a different weight can be assigned in the loss function to each combination of the ground-truth and predicted labels [66]. Here we solely focused on accounting for the imbalance and therefore assigned the weight solely based on the ground-truth label and not on the predicted label.

Table 6 shows the mean and standard deviation over the 13 disasters on different performance measures for the three variations of the model. The macro F1 did not differ much between the three versions, whereas the recall scores of the four classes did. Where the original performance received a better score on the *no damage* class, the model where resampling was applied reached better performance on all three classes that contain damage. This effect can also be seen in the harmonized F1 score, which was higher for the *resample* method. Cost-sensitive learning had less of an effect.

**Table 6.** The results on different performance measures for three versions of the model—*original, resample* and *cost-sensitive*. For each metric the mean and standard deviation (between brackets) are reported and these are calculated over all 13 disasters. *No, Min., Maj., Des.* indicate the *no damage, minor damage, major damage*, and *destroyed* classes, respectively.

| | F1 | | Recall | | | |
|---|---|---|---|---|---|---|
| **Version** | **Harmonized** | **Macro** | **No** | **Min.** | **Maj.** | **Des.** |
| Original | 0.364 (±0.370) | 0.599 (±0.194) | 0.925 (±0.166) | 0.487 (±0.416) | 0.475 (±0.344) | 0.540 (±0.361) |
| Resample | 0.444 (±0.285) | 0.603 (±0.127) | 0.867 (±0.088) | 0.541 (±0.350) | 0.607 (±0.201) | 0.679 (±0.247) |
| Cost-sensitive | 0.365 (±0.298) | 0.548 (±0.202) | 0.869 (±0.103) | 0.497 (±0.383) | 0.454 (±0.332) | 0.596 (±0.359) |

### 5.2. Transferability of Learned Weights to Other Disasters

To assess the performance of the model on a test disaster that was not included in the training set, different configurations of training and testing disasters were experimented with. Two disasters of the set of 13 disasters were chosen as test disasters, namely the Joplin tornado and the Nepal flooding. These were among the best and worst performing disasters, respectively, in the previous experiments. The set of test buildings was kept consistent, namely 10% of one of the two disasters, and different training sets were created according to the following five setups:

Setup 1:  All data of one disaster with the same damage type as the test disaster. This was repeated for all available disasters with this damage type.

Setup 2:  All data of different disasters with the same damage type as the test disaster. Disasters were added progressively, in the order of descending macro F1 (as assessed with setup 1).

Setup 3:  All disasters of setup 2 plus the data of a disaster with a different damage type.

Setup 4:  All data points of a wide mixture of disasters.

Setup 5:  80% of the data of the test disaster.

To assess the change in performance across different disasters, i.e., the transferability of the model, we compared the performance of setup 5 to all the other setups.

Table 7 shows the results of the five setups with the Joplin tornado as the test disaster. For setup 1, where the model was only trained on one disaster, we can see that the performance differed per

training disaster. For three out of four disasters, the performance measures show that the models learned damage-related features that partially transfer to the Joplin tornado. The Moore tornado was the disaster with the highest performance, reaching 80% of the macro F1 compared to being trained on the test disaster itself.

**Table 7.** Performance of models trained on different sets of disasters. The test set is in all cases was 10% of the Joplin tornado. The bottom row indicates the model that was also trained on the Joplin tornado.

| | | Training Disasters | | | | Recall | | | |
| Setup | Names | Types | Regions | Macro F1 | AUC | No dam. | Minor | Major | Destr. |
|---|---|---|---|---|---|---|---|---|---|
| 1 | Moore | tornado (1) | North America (1) | 0.627 | 0.981 | 0.993 | 0.425 | 0.368 | 0.559 |
| 1 | Tuscaloosa | tornado (1) | North America (1) | 0.597 | 0.982 | 0.994 | 0.410 | 0.368 | 0.496 |
| 1 | Michael | hurricane wind (1) | North America (1) | 0.583 | 0.954 | 0.994 | 0.395 | 0.358 | 0.467 |
| 1 | Matthew | hurricane wind (1) | Central America (1) | 0.225 | 0.667 | 0.206 | 0.775 | 0.516 | 0.007 |
| 2 | Moore, Tuscaloosa | tornado (2) | North America (2) | 0.683 | 0.983 | 0.987 | 0.515 | 0.411 | 0.659 |
| 2 | Moore, Tuscaloosa, Michael | tornado (2), hurricane wind (1) | North America (3) | 0.715 | 0.985 | 0.979 | 0.550 | 0.463 | 0.730 |
| 2 | Moore, Tuscaloosa, Michael, Matthew | tornado (2), hurricane wind (2) | North America (3), Central America (1) | 0.733 | 0.984 | 0.976 | 0.585 | 0.589 | 0.700 |
| 3 | Moore, Tuscaloosa, Michael, Harvey | tornado (2), hurricane wind (2), hurricane flood (1) | North America (4), Central America (1) | 0.717 | 0.985 | 0.987 | 0.530 | 0.537 | 0.704 |
| 4 | Michael, Florence, Harvey, Midwest, Guatemala, Matthew, Palu | hurricane wind (2), hurricane flood (2), flood (1), tsunami (1), volcano (1) | North America (4), Central America (2), Asia (1) | 0.485 | 0.963 | 0.989 | 0.075 | 0.116 | 0.807 |
| 5 | Joplin | tornado (1) | North America (1) | 0.787 | 0.990 | 0.961 | 0.705 | 0.516 | 0.907 |

When adding more disasters of the same damage type as the Joplin tornado, i.e., wind, the performance in terms of macro F1 continued to improve (setup 2); see Table 7. When adding a disaster of another type of damage, in this case flood, the macro F1 degraded slightly (setup 3). Training on a mixture of all types of damage led to a drop in macro F1, while still having a high recall for *no damage* and *destroyed*.

From these results, we can conclude that for the case of the Joplin tornado, the models from other domain(s) generalized well, reaching 93% of the macro F1 compared to when trained on the Joplin data. This performance was reached when training on all other wind damage disasters in the dataset. These are the Tuscaloosa tornado, Moore tornado, Michael hurricane, and Matthew hurricane, and are from here on referred to as the *four wind disasters*. To better understand the workings of the model trained on these data, we can look at the confusion matrix, as displayed in Figure 13. The matrix shows that the model learned to recognize damage well and that the main loss in performance came from under-predicting damage.

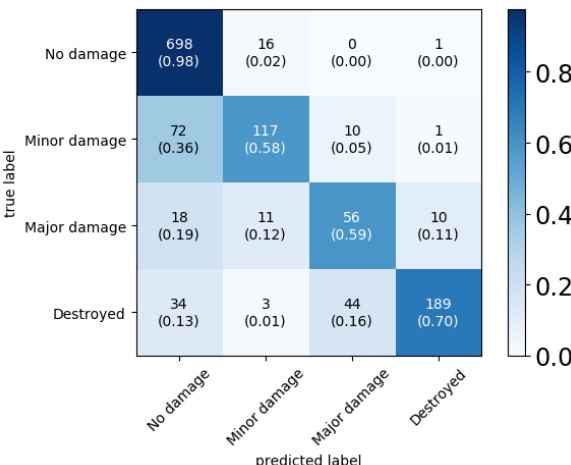

**Figure 13.** Confusion matrix of the model trained on the *four wind disasters* and tested on the Joplin tornado.

The predictions of the model trained on the *four wind disasters* and those of the model trained on the Joplin tornado matched on the largest fraction of buildings. Nonetheless, it is interesting to qualitatively examine buildings that these models did not agree on. The main disagreement was on cases where the model trained on Joplin estimated the damage at a higher level than the model not trained on Joplin; see Figure 14 for two examples. The left example shows a building where the model trained on the *four wind disasters* correctly classified the building as *major damage*, while the model trained on Joplin overestimated the damage as *destroyed*. The border between the two classes here is ambiguous, but it shows that the Joplin model judged the same damage differently. The right building is *destroyed* and was correctly classified by the Joplin model, while the model trained on the *four wind disasters* classifies it as *no damage*. While being wrong, it might be due to poor image quality. The color is clearly different, but the structure appears relatively similar before and after the disaster; thus it might also be a display of damage that does not occur in the *four wind disasters*.

To further understand the performance of the model trained on the *four wind disasters*, we tested it on 100% of the Joplin data and then visualized spatially. Figure 15 shows this visualization for a part of the damaged area. In the figure a *positive* (TP/FP) data point is classified as belonging to the *destroyed* class, and a *negative* (TN/FN) data point to the *not destroyed* class. Here we can see that most incorrect predictions (the orange false negatives and purple false positives) were on the edge of the path of the tornado (the center axis of the image from left to right), which is expected since in those areas the appearances of the damage and their assigned labels are the most ambiguous.

The Nepal flooding is a completely different disaster and thus is a good test case to see if the findings of the Joplin tornado are transferable to other disasters. Table 8 shows the results, from which it can be concluded that the performance differs per disaster when the model is trained on solely one disaster (setup 1). When combining the disasters with flood damage in the training set (setup 2), the performance did not improve compared to the maximum performance of the individual disasters. When adding more disasters of different damage types (setups 3 and 4), the model did not yield better performance than when trained on one disaster either. The highest performance by one disaster was reached when trained on the Midwest flooding and reached 53% of the original macro F1.

In an attempt to better understand the disappointing performance on the Nepal flooding, Figure 16 shows the confusion matrix for the model when trained on the Midwest flooding. The matrix shows that the model only predicted *no damage* and *major damage*. Visual inspection of some of the mispredicted buildings did not show clear indications for the reasons of mispredictions. As an example, Figure 17 shows two mispredicted buildings that quite clearly show their respective classes and thus indicate that, in this case, the model did not learn damage-related features that would be transferable from one flooding disaster to another. Visualizing the predictions made by this model

when tested on 100% of the Nepal flooding, see Figure 18, shows that even clearly damaged areas were not classified correctly.

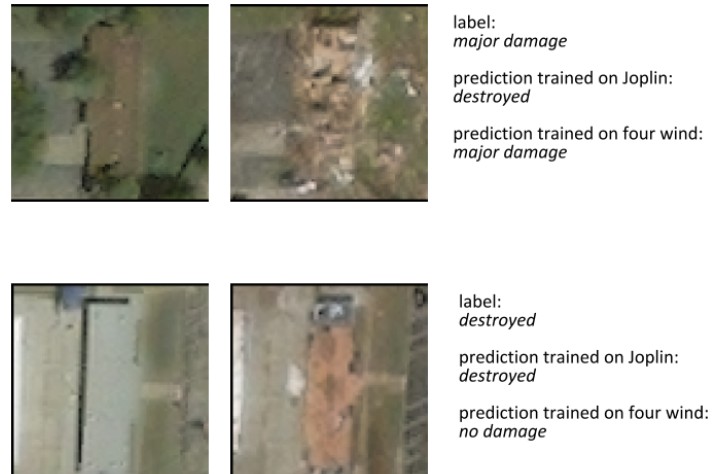

**Figure 14.** Two buildings before and after the Joplin tornado, and the predictions made by the model trained on the Joplin tornado and the model trained on the *four wind disasters*.

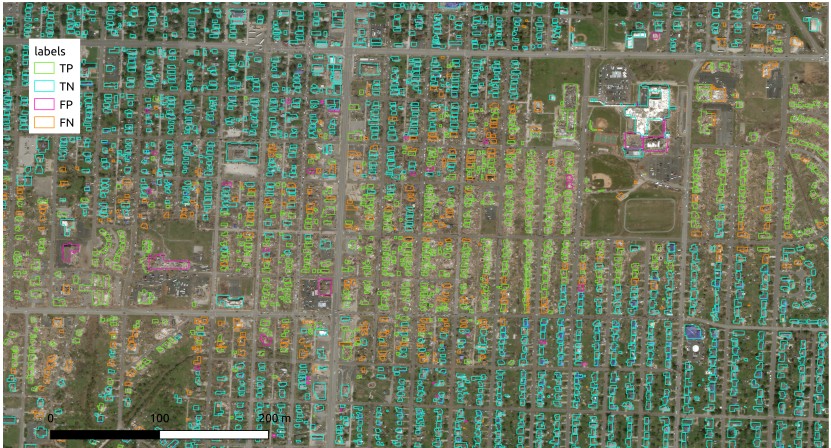

**Figure 15.** The true positive (TP), true negative (TN), false positive (FP), and false negative (FN) predictions on 2395 out of 12,165 buildings of the Joplin tornado by the model trained on a mixture of four disasters with wind damage.

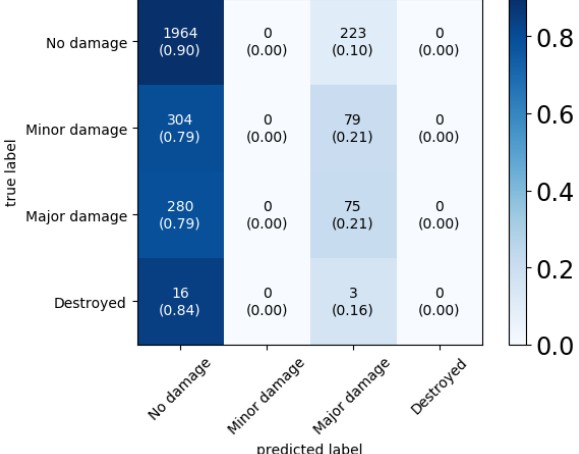

**Figure 16.** Confusion matrix of the model trained on the Midwest flooding and tested on 10% of the Nepal flooding.

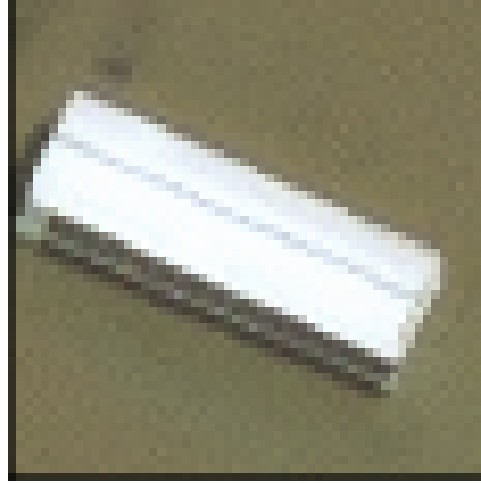
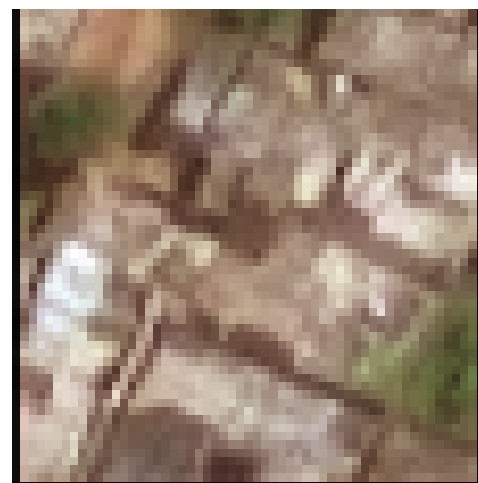

(**a**) Label: *major damage*; prediction: *no damage*　　　(**b**) Label: *no damage*; prediction: *major damage*

**Figure 17.** Two examples of post disaster buildings that were misclassified by the model trained on the Midwest flooding and tested on the Nepal flooding.

**Table 8.** Performance of models trained on different sets of disasters. The test set in all cases was 10% of the Nepal flooding. The bottom row indicates the model that was also trained on the Nepal flooding.

| | Training Disasters | | | | | Recall | | | |
|---|---|---|---|---|---|---|---|---|---|
| Setup Names | Types | Regions | Macro F1 | AUC | No dam. | Minor | Major | Destr. |
| 1 | Midwest | flood (1) | North America (1) | 0.258 | 0.395 | 0.898 | 0.000 | 0.211 | 0.000 |
| 1 | Harvey | hurricane flood (1) | North America (1) | 0.238 | 0.732 | 0.985 | 0.003 | 0.023 | 0.053 |
| 1 | Florence | hurricane flood (1) | North America (1) | 0.173 | 0.480 | 0.382 | 0.013 | 0.504 | 0.000 |
| 2 | Midwest, Harvey | flood (1), hurricane flood (1) | North America (2) | 0.221 | 0.568 | 0.910 | 0.000 | 0.054 | 0.000 |
| 2 | Midwest, Harvey, Florence | flood (1), hurricane flood (2) | North America (3) | 0.217 | 0.711 | 0.995 | 0.000 | 0.008 | 0.000 |
| 3 | Midwest, Harvey, Florence, Palu | flood (1), hurricane flood (1), tsunami (1) | North America (3), Asia (1) | 0.227 | 0.653 | 0.978 | 0.000 | 0.006 | 0.105 |
| 4 | Michael, Florence, Harvey, Midwest, Guatemala, Matthew, Palu | hurricane wind (2), hurricane flood (2), flood (1), tsunami (1), volcano (1) | North America (4), Central America (2), Asia (1) | 0.226 | 0.657 | 0.936 | 0.031 | 0.017 | 0.000 |
| 5 | Nepal | flood (1) | Asia (1) | 0.482 | 0.920 | 0.989 | 0.010 | 0.459 | 0.368 |

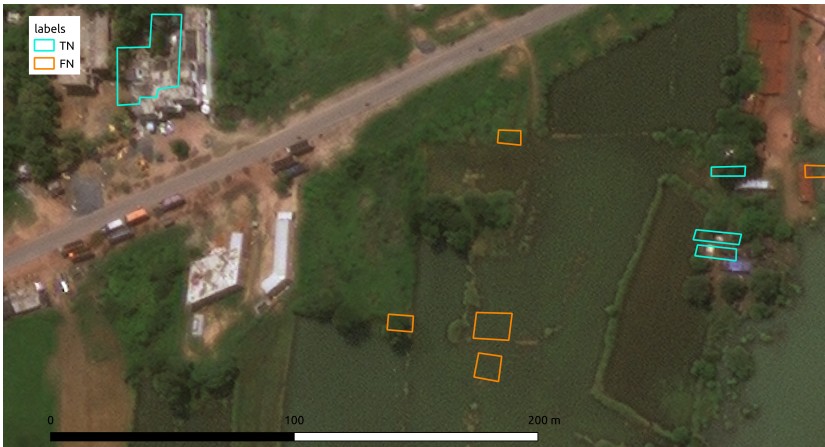

**Figure 18.** The predictions, true negatives (TN), and false negatives (FN) on 9 out of the 29,808 buildings of the Nepal flooding by the model trained on the Midwest flooding.

## 6. Discussion

This study aimed to research the applicability of a CNN-based model in real-time emergency response by testing its performance on 13 disasters caused by natural hazards, with different disaster and image-related characteristics. The results showed that the performance of the model differed significantly across disasters. These differences could not be explained by damage type, geographical location, or satellite parameters, which suggests that other unidentified characteristics of the disaster and/or the images played a more important role. This is surprising, since previous research suggested that image parameters, such as the off-nadir angle, influence the performance [13,67]. Moreover, disaster-specific parameters, such as the number of buildings and hazard type, could have expected to influence the performance. Nevertheless, the fact that these parameters do not have an explaining factor in the performance is also an interesting result. Through the qualitative analysis, it was shown that there is a wide variety of possible explanations of the causes of misclassified buildings. In some cases, the model did not learn the relevant damage features, for example due to lightning, while in some other cases it gave a more realistic classification than the human annotators did. All our results and the conclusions thereof are, of course, conditioned on the xBD dataset, its human labelers, and the quality of the Maxar/DigitalGlobe imagery we used in the experiments. We argue, however, that the used dataset is currently the largest and most advanced that is publicly available, and our conclusions thus bring new insights to the spatial transferability of damage predictions.

The resolution of the classification, i.e., how many distinct classes the model can predict, was studied by comparing the performance in classification between two and four classes. It was shown that the model could indeed classify the middle classes, namely *minor damage* and *major damage*, which was often assumed to be false in previous research. Moreover, when a simpler binary classification is required, the results showed that it is better to still train the model on the multiclass labels and thereafter group them. We hypothesize that this is due to a smaller class imbalance when training on the multiclass labels.

To be of value in real emergency operations, it is also important that the model performs well on a disaster that it has not been trained on. The results showed that the transferability of a trained model to another disaster heavily depends on the test disaster. When setting the Joplin tornado as the test disaster, the transferability was good, reaching 93% of the macro F1 obtained when trained on the same disaster. Concurrently, the other test disaster in this study, the Nepal flooding, showed low transferability, reaching 53% of the original macro F1. What caused this difference between the two test disasters remains an open question. We do not believe there was systematic overfitting, as we have a high performance for the Joplin tornado and low performance for the Nepal floods. The most probable explanation is that the damage characteristics of the Nepal flooding are very specific to that disaster, whereas the Joplin tornado characteristics are also seen in the other disasters. Which set of

training disasters gave the best performance also differed between the two test disasters. However, the specific features associated with damage in a new disaster, as well as the building and environmental characteristics (e.g., water color), are not known a priori, and thus it is not known which of the individual disasters would result in the best performance. Thus, it seems to be a reasonable choice to still utilize the model trained on all disasters of the same damage type, since this gives a similar level of performance to when trained on the best individual disaster while providing more certainty on the level of this performance.

## 7. Conclusions and Future Work

In conclusion, we systematically evaluated the performance of a CNN-based model in a wide variety of operational conditions, which has never been done so extensively before. When training and testing on the same disaster, as well as when leaving the test disaster out of the training set, it was found that the performance heavily differed per test disaster. These differences could not be explained in terms of disaster- or image-specific characteristics, such as the damage type, geographical location, and satellite parameters. This suggests that great care is imperative when using such a model on a new disaster and that manual validation of (part of) the output might be necessary.

At the same time, the results showed that it is possible to reach a high performance on a new, unseen disaster when training on different disasters with the same damage type. This constitutes the first evidence ever that a CNN-based model can perform sufficiently well for emergency response operations, where model (re-)training is not an option. When operationalizing our approach, it is important to create an end-to-end deep learning task pipeline [55]. Obtaining, storing, processing, and modeling the satellite imagery of the impacted area (as usually released by the International Charter: Space and Major Disasters) in combination with the pre-disaster imagery should be done fast and efficiently. More research is required to better understand the limits of applicability of such an approach, which were, e.g., already encountered in the case of the Nepal flooding. This can be done by testing on a larger set of data, especially in the scenario where the test disaster type is excluded from the training set, to assess if this shows any relations between performance and disaster type-specific parameters. This could for example show if it would be more beneficial to train the model on disasters that are geographically similar to the test disaster, a suggestion that was made by Nex et al. [13] but which could not be tested with the data used in this research. If this would be the case, adaptations to the generic pipeline should be made, creating area-specific pipelines. Moreover, it should be examined whether pipelines specific to disaster type should be made. For example, from the qualitative analysis it might be suspected that for floods a larger bounding box around the building might prove to be useful. Lastly, a more elaborate qualitative analysis might provide useful insights into the variation of data from different disasters and the nature of the misclassifications.

This study focused on the modeling aspects of automated building damage assessment that are essential for the implementation of the model into emergency response operations. The results in terms of performance measures are promising, but their real added value can only be assessed by end-users. Moreover, the information produced by this model has to be combined with other information, such as building function and vulnerability information, to make a complete damage assessment. In this way, for example the damage to buildings used for public services can be singled out rapidly. Lastly, the implementation into the emergency response operations will also require to better understand the model's position and scope in the current damage assessment processes, e.g., when is the information needed, which detail of damage classes should be provided, and to which other data source(s) should the assessment be connected. Participatory design with the envisioned end-users (e.g., the Global Shelter Cluster) will likely be necessary.

**Supplementary Materials:** The code of the model and all of the experiments are available online at https://github.com/rodekruis/caladrius.

**Author Contributions:** T.V. designed and conducted the experiments and wrote the first draft. J.M., M.v.d.H., and J.L. guided this process and helped with the writing of the paper. All authors have read and agreed to the published version of the manuscript.

**Funding:** This research received no external funding.

**Acknowledgments:** This research would not have been possible without all the contributions of the amazing team of volunteers who were involved at 510, an initiative of the Netherlands Red Cross. We also thank the creators of the xBD dataset for producing the openly available and high-quality data.

**Conflicts of Interest:** The authors declare no conflict of interest.

## Abbreviations

The following abbreviations are used in this manuscript:

CNN   convolutional neural network
DNA   damage and need assessment
UAV   unmanned aerial vehicle
GIS   geographic information system
HDX   Humanitarian Data Exchange
RGB   red, green, blue
ReLU  rectified linear unit
TP    true positive
TN    true negative
FP    false positive
FN    false negative
AUC   area under curve
GSD   ground sample distance
F1    harmonic mean of precision and recall

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
