# Peer review of "Multi-Hazard and Spatial Transferability of a CNN for Automated Building Damage Assessment"

_remotesensing, doi:10.3390/rs12172839_

Round 1

Reviewer 1 Report

Please find the comments in the attachment.

Reviewer 2 Report

Please look at the attached word file.

Round 2

Reviewer 1 Report

Dear Authors

It is good to see your revised manuscript. Most of the comments were well answered and changes appropriately considered. But still, there are some critical issues that following them will help your article to be acceptable in publishing in this journal. Firstly, I am not sure by having such a high accuracy just by CNN and fundamental optimization about the accuracy of the data and applicability of the method. It has to be clarified in the next response letter. Your data is not big and surely there must be somewhere overfitting. Indeed, there is no structural type and damage scales considered. For this purpose you can follow the EMS-98 scales and clarify your method for which type of buildings were. Are all of them RC buildings? 

As I said last time, beside intensive works and presentation of the result of your work, it does not have novelty, and the problem statement is very poor, regarding the standard of this journal. The CNN is already a technique where you implement it.

Therefore, I suggest you make the literature mature and give a brief overview of similar methods before and after damage. For this purpose, you can use the good works I have shared with you in the 1st round.

I would recommend short the title of tables and figures e.g., Fig. 1 or Table 2, that have about four lines description where you can use it in the text body.
It would be good to provide a flowchart (somehow the same as fig.1) that how the emergency services can benefit from your methodology and the parts and process involved. It would be good if you could graphically provide it in the conclusion section.  

There are still different shape of tables which are not following each other.

Author Response

Dear reviewer, 

Thank you for your feedback. We have done our best to improve and/or clarify your points of attention. 

We have included a section in the introduction with a flow chart of how our developed model fits in the whole pipeline after a disaster strikes. This also connects to our novelty, since it is of utmost importance that a developed model actually works within this set flow chart. To our knowledge, this hasn’t been studied yet; no studies have been done on multi-hazard and spatial transferability, not even when labeled training data is assumed. Besides the lack of experiments with multi-hazard and multi-geolocations,  there is generally a lack of studies where it is assumed that labeled training data is not available, which is a more realistic scenario. We hope we have now clearly communicated this novelty in our text.

We have revised the tables so that they now look more uniform. Naturally, they cannot all look exactly the same because the information they convey is different, but the basic formatting principles have been the same for all.  Tables 1, 7, and 8 have alternating white and grey backgrounds for the lines for clarity, but it is OK for us if also those tables will be made single-colored by the copy editor.

Regarding the data and the overfitting, we have added a piece of text in the discussion that these results are solely based on the xBD dataset and thus we cannot guarantee no overfitting to this particular dataset. Nevertheless, we believe this data was the best openly available data to use and we don't believe our model is heavily overfitting. We can understand that the reported accuracies seem very high, but as emphasized in the paper accuracy is not a very good measure to look at due to the class imbalance. For some of the disasters, 98% of the buildings are not-destroyed, and therefore the accuracy should be at least 0.98 for the model to have any predictive value. We have compared train, validation, and test performance measures and in those, no clear signs of overfitting are seen. This is also highlighted by the difference in performance when testing on the Nepal and Joplin tornado, where if overfitting would indeed be taking place, performance for both disasters would be expected to be bad. 

For the literature overview, we have taken a broader view and explained why our research focuses on predicting damage after a disaster struck with solely using remote sensed data. Thus, we have included works on forecasting the damage before a disaster strikes, and works that use other geospatial data (as two of your suggested papers do). Moreover, we have extended our discussion of different methods and why a CNN-based model seemed the most promising method for our purposes. 

We hope we have hereby addressed all your comments and questions. 

Reviewer 2 Report

The authors addressed all the comments, recommendations, and significantly improved the manuscript. The only remaining point is that the figures' captions must be revised according to the image labels in the figures, the current version is confusing (e.g. figure 4). 

Author Response

Thank you. We have clarified the captions and made some additional changes based on the other reviewer's comments.